

# Composite-boson formalism applied to strongly bound fermion pairs in a one-dimensional trap

Martín D. Jiménez[1], Eloisa Cuestas[1,2], Ana P. Majtey[1] and Cecilia Cormick[1⋆]

**1** Instituto de Física Enrique Gaviola, CONICET and Universidad Nacional de Córdoba, Ciudad Universitaria, X5016LAE, Córdoba, Argentina
**2** Quantum Systems Unit, Okinawa Institute of Science and Technology Graduate University, Onna, Okinawa 904-0495, Japan

⋆ cecilia.cormick@unc.edu.ar

## Abstract

We analyze a system of fermions in a one dimensional harmonic trap with attractive delta-interactions between different fermions species, as an approximate description of experiments involving atomic dimers. We solve the problem of two fermion pairs numerically using the so-called "coboson formalism" as an alternative to techniques which are based on the single-particle basis. This allows us to explore the strongly bound regime, approaching the limit of infinite attraction in which the composite particles behave as hard-core bosons. Our procedure is computationally inexpensive and illustrates how the coboson toolbox is useful for ultracold atom systems even in absence of condensation.

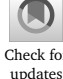
# 1  Introduction

The possibility to engineer atomic and molecular many-body systems by controlling and assembling simpler components has made enormous progress thanks to Feshbach resonances. In this way, molecular Bose-Einstein condensates have been formed starting from ultracold atomic gases [1, 2]. Similar setups have been used for the controlled observation of relevant phenomena in statistical physics such as Wigner crystals [3] and the BEC-BCS crossover [4, 5]. Within the field of ultracold Fermi gases, one-dimensional systems are known to exhibit very peculiar properties [6]. In particular, strongly bound fermion pairs reach a limit in which they behave as hard-core bosons, which in turn are related with non-interacting fermion models [7].

We consider a one-dimensional scenario, with fermions of two different kinds in a harmonic trap and an attractive contact interaction leading to fermion pairing. The first steps towards the exact solution of the one-dimensional Fermi gas with contact interactions in a ring are due to Gaudin and Yang in 1967 [8, 9]. For the trapped case most of the analytical work focuses on the strongly repulsive case, see [10] and references therein. Numerical approaches for this system include multiconfigurational time-dependent Hartree method [11], quantum diffusion Montecarlo [12], density matrix renormalization group [13] and a variety of quantum-chemical treatments such as coupled-cluster methods [14], among others. The vast body of literature in this field has been reviewed for instance in [6, 15].

Even though much effort has been devoted to this system, the usual numerical treatment takes as a basis the harmonic oscillator eigenstates, making computations very costly for strong attraction [14, 16–20]. Alternative procedures which are more efficient for strong attraction have been proposed in [21, 22]. Here, as a different approach, we tackle the problem of two pairs with two fermions each in the context of coboson theory [23, 24]. This theoretical framework, originally developed for excitons in semiconductors [23–25], has by now been applied to a variety of systems, including Bose-Einstein condensates [26], superconductors [27, 28] and Feshbach molecules [29].

A very useful simplification often encountered in this treatment is the so-called coboson ansatz, which is analogous to a condensate formed by composite bosons and is the canonical-ensemble counterpart of the BCS ansatz [23, 28]. Using tools from the coboson formalism, we show that the coboson ansatz does not provide a good approximation of the true ground state for the case of two pairs in the limit of strong interaction. This is to be expected in the light of previous results [30, 31] and also because the limit of infinitely bound pairs corresponds to hard-core bosons which are known to form only a quasi-condensate in 1D traps [32–34]. However, the coboson formalism also provides tools to describe the state beyond the coboson ansatz [27, 28]. We thus develop a representation of the problem in the coboson basis, i.e. in terms of the eigenstates of one pair of interacting fermions in the trap.

This basis is specially convenient and expected to work better for the regime of strong attraction, which is difficult to address numerically (see for instance Ref. [16]) and has been not studied exhaustively as the repulsive regime [6, 15]. In this respect, our method is related with the perturbative approach in [35]. The case of two pairs is of particular relevance within

the coboson formalism, however, the method we propose can be extended to larger systems. The motivation of our work can then be stated as i) to show that even if the coboson ansatz fails the correct ground state for this system can be recovered using the complete toolbox of the coboson formalism ii) to show that the two-body coboson basis is useful in the strongly attractive limit where the single-particle basis is not convenient.

Besides the numerical convenience of using the coboson basis, studying this system within the coboson formalism leads to semi-analytical reliable results that can provide a safe ground to quantify the fractional statistics [36,37] of the one-dimensional Fermi gas [38–40]. This is a good starting point to analyze the relationship between anyonic statistics and the entanglement of the constituent particles of the composite boson, which has been pointed out to be the key to understand composite effects and ideal bosonic behavior [41–43].

The basic steps of our procedure to tackle the problem of two trapped fermion pairs are the same as in [31] and are as follows:

1. We solve the problem of a pair of interacting fermions in the trap. The operators $B_n^\dagger$ that create each single-pair eigenstate, and the corresponding energies $E_n$, will be the starting point of the treatment. We truncate the basis considering the states with the lowest energies, up to some quantum number $n_{\max}$.

2. From the single-pair basis operators $B_n^\dagger$ we form the two-coboson basis generated by the action on the vacuum of operators of the kind $B_n^\dagger B_m^\dagger$.

3. We calculate the form of the Hamiltonian in this truncated coboson basis.

4. Solving the corresponding generalized eigenvalue problem, we estimate the ground state for two pairs and analyze its properties.

This method allows us to interpolate from the interaction strengths for which the single-particle basis is suitable [17–20], all the way to very strongly bound pairs approaching the limit of hard-core bosons. Using coboson-theory tools combined with Taylor expansions, we calculate several quantities of interest, including the energy and two-particle correlators.

The work is presented as follows: in Sec. 2 we review how to write the problem in the coboson framework. Section 3 is devoted to analytical considerations for infinite attraction. In Sec. 4 we discuss our numerical results. A summary and conclusions are given in Sec. 5. Finally, several appendices with detailed calculations are included.

## 2 The procedure, step by step

### 2.1 Single-pair solution

For definiteness we will assume that both fermion kinds, which we call $a$ and $b$, have the same mass, and that the creation and annihilation operators corresponding to different fermion species commute (this last choice does not affect the final results). We also assume that the trapping potential is the same for both species.

The first step requires the solution of the single-pair problem, with a Hamiltonian given by:

$$H_1 = \sum_{\alpha=a,b} \left( \frac{p_\alpha^2}{2m} + \frac{m\omega^2 x_\alpha^2}{2} \right) - \gamma\,\delta(x_a - x_b). \tag{1}$$

This problem can be solved by separation of the center-of-mass and relative variables. The center-of-mass solution is given by the harmonic oscillator eigenfunctions corresponding to mass $2m$. The relative motion has been solved in the general case in Refs. [44, 45] but for

simplicity we focus only on strongly bound pairs, so that the relative motion has a wavefunction of the form of an exponential,

$$\psi_r(x_r) \simeq \sqrt{\lambda}\, e^{-\lambda|x_r|}, \tag{2}$$

and the energy associated with the relative motion can be approximated by:

$$E_\gamma = -\frac{\hbar^2\lambda^2}{m}, \quad \lambda \simeq \frac{m\gamma}{2\hbar^2}. \tag{3}$$

In this regime, the single-pair eigenfunctions are then approximately of the form:

$$\psi_n(x_a, x_b) \simeq \varphi_n\left(\frac{x_a + x_b}{2}\right)\sqrt{\lambda}\, e^{-\lambda|x_a - x_b|}, \tag{4}$$

where $\varphi_n$ are the harmonic oscillator eigenfunctions for a particle of mass $2m$. The corresponding energies are:

$$E_n = \hbar\omega\left(n + \frac{1}{2}\right) + E_\gamma. \tag{5}$$

From these solutions, we define the coboson creation operators $B_n^\dagger$ such that:

$$|\tilde{n}\rangle = B_n^\dagger|\nu\rangle, \tag{6}$$

where $|\tilde{n}\rangle$ is the $n$-th single-pair eigenstate, and $|\nu\rangle$ is the vacuum. In particular, the coboson operators $B_n^\dagger$ can be written in terms of field operators as:

$$B_n^\dagger \simeq \int \mathrm{d}x_a \mathrm{d}x_b\, \psi_n(x_a, x_b)\Psi_a^\dagger(x_a)\Psi_b^\dagger(x_b). \tag{7}$$

For consistency, neglecting states where the internal motion is excited implies also a truncation in the center-of-mass states, so that the basis includes all single-pair eigenstates up to a certain energy cutoff. In particular, we keep only states where the index $n$ associated with the center-of-mass motion is such that the excited internal states are well above the energy scales considered, i.e.:

$$n \ll \frac{|E_\gamma|}{\hbar\omega} = (\lambda x_\omega)^2. \tag{8}$$

For convenience here we have defined a spatial scale $x_\omega$ associated with the harmonic oscillator,

$$x_\omega = \sqrt{\frac{\hbar}{m\omega}}. \tag{9}$$

The inequality in Eq. (8) stresses once more the fact that our restricted basis is only appropriate for strong attraction, when the size of each bound pair is very small compared with the spatial scale of the trap and thus $\lambda x_\omega$ is large. It is also important to note that since Eq. (2) and therefore Eq. (4) are valid for $\lambda x_\omega \gtrsim 5$ all of our results rely on this condition [46].

## 2.2 Basis for two pairs

From the set of states corresponding to the lowest energies of the single-pair Hamiltonian, one can form states of the form:

$$|\tilde{n}\tilde{m}\rangle = B_n^\dagger B_m^\dagger|\nu\rangle, \tag{10}$$

with $n \leq m$ (we note that the coboson creation operators commute) and $|\nu\rangle$ the vacuum. Because of the fermionic character of the constituent particles, states generated in this form

are neither normalized nor orthogonal [23]. We truncate this two-pair basis with the condition $n + m \leq n_{\max}$, and then approximate the ground state in the form:

$$|GS\rangle = \sum_{m \leq n} c_{m,n} |\tilde{n}\tilde{m}\rangle . \qquad (11)$$

An often useful approximation for the ground state of dilute systems of $N$ pairs with short-range interactions is given by what we call the "coboson ansatz" [23]. This corresponds to the state obtained from the repeated application on the vacuum of the operator $B_0$ that creates a single pair in its ground state:

$$|N\rangle = \frac{(B_0^\dagger)^N}{\sqrt{N! \chi_N}} |v\rangle , \qquad (12)$$

where $\chi_N$ is a normalization constant. However, this can only provide a good approximation of the true ground state in systems which are expected to exhibit condensation at zero temperature. This is not the case in the problem we analyze [30, 31, 33, 34]. In order to quantify the quality of the approximation, we study the fidelity $\mathcal{F}$ between the true ground state for two pairs, $|GS\rangle$, and the coboson ansatz:

$$\mathcal{F} = \frac{|\langle GS|(B_0^\dagger)^2|v\rangle|^2}{\langle v|B_0^2 (B_0^\dagger)^2|v\rangle} , \qquad (13)$$

where the true ground state $|GS\rangle$ is approximated numerically using the coboson basis given in Eq. (10) for two-pairs ($N = 2$).

Even if the coboson ansatz is not a good approximation, one can still compute the ground state by means of the coboson formalism. In order to do this, we will work with the space generated by the coboson operators as in Eq. (10). First, we compute all overlaps between the relevant states from the expression:

$$S_{kl,mn} = \langle v|B_k B_l B_m^\dagger B_n^\dagger|v\rangle = \delta_{ml}\delta_{kn} + \delta_{nl}\delta_{km} - \left[ \langle \tilde{k}| \otimes \langle \tilde{l}|X_a|\tilde{m}\rangle \otimes |\tilde{n}\rangle + \langle \tilde{k}| \otimes \langle \tilde{l}|X_b|\tilde{m}\rangle \otimes |\tilde{n}\rangle \right]. \qquad (14)$$

Here $X_\alpha$ with $\alpha = a, b$ is an operator that exchanges the states of the two fermions of kind $\alpha$, and it acts on a fictitious space where fermions of equal kind are treated as distinguishable. Since our goal is to find the ground state, instead of building an orthonormal basis, we keep the overlap matrix $S$ to solve the corresponding generalized eigenvalue problem.

The matrix $S$ can be calculated following different strategies. In the coboson literature [23], the overlaps are evaluated in terms of matrix elements of the change of basis between single-pair eigenstates and the separable single-fermion basis. However, this procedure can be numerically costly and lead to large errors when many coefficients are non-negligible and no analytical expression exists for the sums required. Thus, we resort to a different form of evaluation. Plugging the explicit form of the operators $B_n^\dagger$ given by Eq. (7) in all formulas, and using (anti)commutators, we can obtain an expression for the elements of the overlap matrix as:

$$S_{mn,jk} \simeq \left[ \delta_{mj}\delta_{nk} - \lambda^2 \int dx \, dy_1 \, dy_2 \, dy_3 \, dy_4 \, \delta(y_1 + y_2 - y_3 - y_4) \right.$$

$$\left. \times \varphi_m(x) \varphi_n\left(x + y_3 - \frac{y_1 + y_2}{2}\right) \varphi_j\left(x + \frac{y_3 - y_1}{2}\right) \varphi_k\left(x + \frac{y_3 - y_2}{2}\right) e^{-\lambda \sum_l |y_l|} \right]$$

$$+ \text{ same with } j \leftrightarrow k. \qquad (15)$$

Since we are interested in the case of strong attraction, the factors of the form $e^{-\lambda|y_l|}$ allow us to perform a Taylor expansion in $1/(x_\omega \lambda)$ for the harmonic oscillator functions $\varphi_n$. This

is possible given the truncation of our basis in Eq. (8), which implies that the spatial scale associated with the center of mass is much longer than the pair size $\lambda^{-1}$. In this form one can find approximate expressions for $S$ from a lengthy but straightforward evaluation of spatial integrals. This procedure is explained in detail in Appendix A.

### 2.3 Construction of the Hamiltonian

We now need to compute the Hamiltonian in the coboson basis. The Hamiltonian can be split in two parts, corresponding to the non-interacting terms and the interactions. The interaction part is quartic and can be written in terms of field operators as:

$$H_{\text{int}} = -\gamma \int dx\, \Psi_a^\dagger(x)\Psi_b^\dagger(x)\Psi_a(x)\Psi_b(x). \tag{16}$$

The Hamiltonian matrix elements in the coboson basis can be obtained from the expression:

$$\langle v|B_k B_l H B_m^\dagger B_n^\dagger|v\rangle = (E_n + E_m)S_{kl,mn} + \langle v|B_k B_l\big[[H_{\text{int}}, B_m^\dagger], B_n^\dagger\big]|v\rangle, \tag{17}$$

which is just a rewriting of the formulas in [23]. Notice that when using the coboson formalism the one-body term which contains the kinetic energy and trap potential is absorbed by quantities that were calculated when solving the single-pair case (first term on the right-hand-side in the above equation). In a similar spirit as for the calculation of the overlap matrix $S$, instead of following the standard expressions in [23] we estimate the Hamiltonian elements using a Taylor expansion of spatial integrals.

In particular, the last line of Eq. (17) can be written as:

$$\langle v|B_m B_n\big[[H_{\text{int}}, B_j^\dagger], B_k^\dagger\big]|v\rangle \simeq$$

$$\gamma\lambda^2\Bigg[\int dx\,dy\,dy'\, e^{-\lambda(|y|+|y'|+|y-y'|)}\varphi_m(x)\varphi_n\left(x+y'-\frac{y}{2}\right)\varphi_j\left(x+\frac{y'-y}{2}\right)\varphi_k\left(x+\frac{y'}{2}\right)$$

$$-\int dx\,dx'\,dy\,dy'\, e^{-2\lambda(|y|+|y'|)}\varphi_m(x)\varphi_n(x')\varphi_j(x)\varphi_k(x')\delta\left(x-x'+\frac{y+y'}{2}\right)\Bigg]$$

$$+ \text{ same with } n \leftrightarrow m + \text{ same with } j \leftrightarrow k + \text{ same with } \{j,k\} \leftrightarrow \{m,n\}. \tag{18}$$

The details of the procedure involving the Taylor expansion of the Hamiltonian elements are also provided in Appendix A.

## 3 Analytical considerations for infinite attraction

Before presenting the results of our numerical approach, we note that the case of infinite attraction can be solved exactly. In this limit, fermions of different species are so strongly bound that they behave as point-like hard-core bosons of mass $2m$, and the problem can be solved by means of fermionization [7]. According to this procedure, one must first consider the ground state of two identical non-interacting fermions of mass $2m$ in the trap. This state is given by:

$$\psi_{2\text{f}}(x_1, x_2) = \frac{2m\omega}{\hbar\sqrt{\pi}}e^{-m\omega(x_1^2+x_2^2)/\hbar}(x_1-x_2), \tag{19}$$

and corresponds to the antisymmetric combination of having one fermion in the trap ground state and another in the first excited state. Then, one obtains the wavefunction of the hard-core bosons as the symmetrized form of the previous expression, i.e.:

$$\psi_{\text{hc}}(x_1, x_2) = \frac{2m\omega}{\hbar\sqrt{\pi}}e^{-m\omega(x_1^2+x_2^2)/\hbar}|x_1-x_2|, \tag{20}$$

where the subindex "hc" stands for "hard-core".

From these expressions we can calculate all properties of the ground state for $\lambda \to \infty$. For instance, the asymptotic ground-state energy, excluding the binding energy $E_\gamma$ of each pair, is found to be given by the sum of the two lowest energies of the harmonic oscillator. Thus, the total ground-state energy for very large $\lambda$ is approximately $2E_\gamma + 2\hbar\omega$. We can define an effective interaction energy between pairs as $\Delta E = E_2 - 2E_1$, where $E_N$ is the ground-state energy of $N = 1, 2$ pairs. Considering that a single pair has a ground-state energy of $E_\gamma + \hbar\omega/2$, we then obtain an effective interaction energy which for very large attraction approaches $\Delta E = \hbar\omega$.

Using the ground-state wavefunction as expressed above, one can also analytically calculate the fidelity between the true ground state and the coboson ansatz for infinite attraction. We find an asymptotic fidelity of $\mathcal{F}_\infty = 2/\pi \simeq 0.64$, which is lower than the one obtained in the same regime for two fermion pairs in translationally invariant models [30, 31].

Following the same lines, one can find the joint density of composite particles at positions $x$ and $x'$ for the limit of infinite attraction. This is of the form:

$$\mathcal{D}_{\text{hc}}(x, x') = \frac{8}{\pi^2} \left( \frac{x - x'}{x_\omega} \right)^2 e^{-2(x'^2 + x^2)/x_\omega^2}. \tag{21}$$

One can also write down the conditional probability $\mathcal{P}(x'|x)$ of finding a composite point-like particle at position $x'$ provided that another one was found at position $x$:

$$\mathcal{P}_{\text{hc}}(x'|x) = \frac{1}{x_\omega} \sqrt{\frac{2}{\pi}} \frac{(x - x')^2}{x^2 + x_\omega^2/4} e^{-2(x'/x_\omega)^2}. \tag{22}$$

Furthermore, one can calculate the asymptotic values of the coefficients in the expansion of the ground state in the coboson basis, Eqs. (10-11), obtaining for $\lambda \to \infty$:

$$c_{mn}^{(\infty)} = -(2 - \delta_{mn}) \frac{(-1)^{(m-n)/2}}{\sqrt{m!n!}} \sqrt{\frac{1}{\pi}} \frac{(m+n)!}{(m/2 + n/2)!} \frac{1}{2^{m+n}} \frac{1}{m + n - 1}. \tag{23}$$

This expression is valid for even and nonzero $n + m$, and here $\delta_{mn}$ is the Kronecker delta. For symmetry reasons the coefficients $c_{mn}$ vanish for odd $n + m$, and for $n = m = 0$ we find:

$$c_{00}^{(\infty)} = \sqrt{\frac{1}{\pi}}. \tag{24}$$

Since the coboson ansatz corresponds to the repeated application of the coboson operator $B_0^\dagger$, and for $\lambda \to \infty$ the wavefunctions associated with the different $B_m^\dagger$ become orthogonal, the asymptotic value of $c_{00}$ determines the asymptotic fidelity between the correct ground state and the coboson ansatz. The additional factor $\sqrt{2}$ in the fidelity comes from the definition of the coboson basis in Eq. (10), which does not include a prefactor $1/\sqrt{2}$ for $m = n$.

Before tackling the numerical treatment of the problem for strong but finite attraction, we note that also the limit of infinitesimal attraction can be treated analytically. For $\gamma = 0$, the ground state of the system is separable, with the two lowest oscillator levels occupied for both kinds of fermions. Then, the energy $\Delta E$ approaches $2\hbar\omega$. It is very important to notice that in this separable limit, the coboson normalization factor $\chi_2$ in Eq.(12) vanishes, and thus the coboson ansatz is not defined for $\gamma = 0$. Nevertheless, using perturbation theory together with analytical results for the Schmidt coefficients [46] one can calculate the limit value of the fidelity between the true ground state and the coboson ansatz, and find that as the attractive interaction strength approaches zero, $\mathcal{F}$ approaches a value of approximately 0.37. Indeed, for $\gamma \sim 0$ we obtain $\chi_2 \sim 0.342\,\theta^2$ and $\mathcal{F} \sim \theta^2/8\chi_2$ with $\theta \sim \gamma/\sqrt{2\pi}\hbar\omega x_\omega$. We note, however, that the weakly bound case is not within the scope of our present study, and it has been extensively analyzed before [17–20].

# 4 Numerical study of the ground state for strong attraction

In the following we perform a numerical study of the ground state according to the procedure outlined in Sec. 2. A delicate point in the calculation is the choice of the number of basis states. A very small number leads to a poor description of the system, whereas for a very large number it becomes unjustified to leave out the excited states of the relative motion, and it can also lead to numerical problems if the overlap matrix becomes worse conditioned. As a compromise, we choose the maximum center-of-mass energy included in our description to grow linearly with $\lambda$.

In Fig. 1 (a) we show our results for the interaction energy $\Delta E = E_2 - 2E_1$ using a Taylor expansion for the calculation of both the overlap and the Hamiltonian matrices. We also plot in Fig. 1 (b) the fidelity $\mathcal{F}$ between the ansatz and the true ground state as a function of $\lambda$ when choosing the energy in the truncated basis to be given by $n_{\max} = \lambda x_\omega$. We note that our results show reasonable agreement with the known behaviour for infinite attraction. Notice that the difference between the numerical $\Delta E$ obtained for $\lambda x_\omega \simeq 200$ and the asymptotic value $\hbar\omega$ presented in Fig. 1 (a) is of about 4%, whereas the binding energy for this case is so large that $\Delta E$ is five orders of magnitude smaller than the total energy.

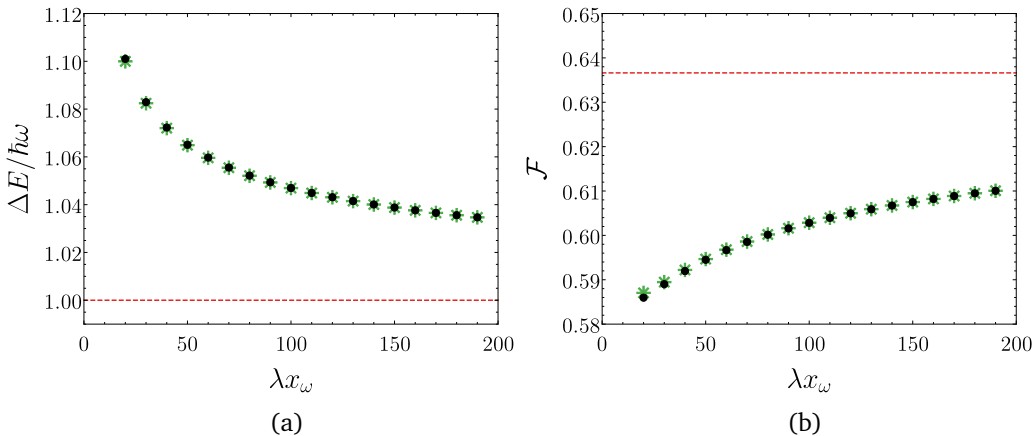

(a)             (b)

Figure 1: a) Energy for two pairs, excluding the trivial contribution equal to twice the single-pair energy, as a function of $\lambda$. b) Fidelity between the coboson ansatz and the numerically found ground state as a function of $\lambda$. The results are obtained from the lowest non-trivial order of the Taylor expansion (green stars) and the next non-zero higher-order corrections (black circles) as reported in Appendix A. The horizontal dashed red lines indicate the asymptotic values for $\lambda \to \infty$.

As can be seen in the comparison provided in Fig. 2, for $\lambda x_\omega = 30$ the coefficients $c_{m,n}$ of the ground state in the form of Eq. (11) are already very close to the ones obtained from the hard-core boson limit given in Eqs. (23-24). This also hints at a procedure to perform approximate computations more efficiently: instead of taking the full basis as in Fig. 2, one can use a truncation inspired by the asymptotic values of the coefficients in Eqs. (23-24). One can also directly approximate the state by taking the coboson basis in Eq. (10) to be a function of $\lambda$ but the coefficients in this basis to be given by the asymptotic values, which gives a fast and compact approximation for the ground state. Indeed, the ground state found numerically for $\lambda x_\omega = 30$ has a fidelity of 0.993 with the state obtained taking the asymptotic values of the coefficients and truncating the basis in the same form.

From the numerical solution of the problem one can characterize the ground state through several key properties. In particular, in Fig. 3 (a) we illustrate the spatial correlations between

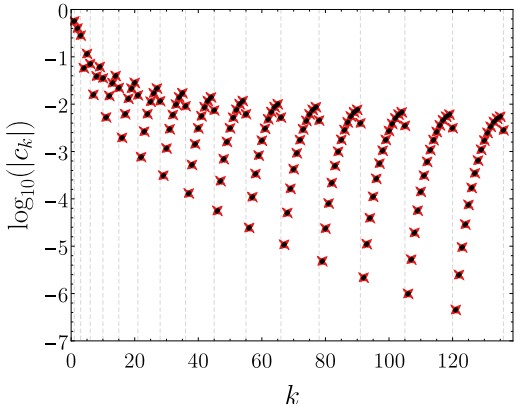

Figure 2: Coefficients in the coboson decomposition from the numerical resolution of the problem based on a Taylor expansion for $\lambda x_\omega = 30$, in black circles. The index $k$ here refers to a particular ordering of the $m, n$ coefficients using a single label. For comparison we show the values according to the asymptotic expression in Eqs. (23-24) as red four-pointed stars, which overlap with the numerical resuls within the size of the symbols. The basis was truncated with $n_{\max} = \lambda x_\omega$. The coefficients in the plot are normalized taking $c^t S c = 1$. The vertical dashed light-gray lines delimitate sections of the basis containing states $B_m^\dagger B_n^\dagger |v\rangle$ with a fixed value of $m + n$.

fermions of equal kind through the joint density distribution

$$\mathcal{D}_{aa}(x, x') = \langle \psi | \Psi_a^\dagger(x') \Psi_a^\dagger(x) \Psi_a(x) \Psi_a(x') | \psi \rangle, \tag{25}$$

evaluated for the case $\lambda x_\omega = 30$. The details of the calculation are provided in Appendix B. This plot displays clear signatures of Pauli exclusion as a sharp diagonal feature. Two identical fermions are most likely found apart from each other at a distance which is set by the spatial scale of the harmonic trap.

For comparison, Fig. 3 (b) displays the joint density for fermions of different kinds:

$$\mathcal{D}_{ab}(x, x') = \langle \psi | \Psi_b^\dagger(x') \Psi_a^\dagger(x) \Psi_a(x) \Psi_b(x') | \psi \rangle. \tag{26}$$

This plot exhibits a strong diagonal correlation corresponding to particles that form a bound pair, with additional much broader peaks corresponding to particles belonging to different pairs. The calculation of $\mathcal{D}_{ab}$ is explained in Appendix C.

Another quantity that reflects the spatial correlations present in the ground state is the conditional probability $\mathcal{P}_{aa}(x|0)$ to find one fermion of kind $a$ at position $x$ given that another identical fermion was found at the origin. This function is plotted in Fig. 4 (a), for the numerical solution with $\lambda x_\omega = 30$. For comparison we also show the conditional probability $\mathcal{P}_{aa}(x|0)$ obtained from the hard-core limit of $\lambda \to \infty$ and from the coboson ansatz of Eq. (12) evaluated for $\lambda x_\omega = 30$. The corresponding formulas are given in Appendix B. The plots show qualitative agreement between the numerical results and the point-like hard-core boson limit, in sharp contrast with the coboson ansatz in its standard form. Indeed, the form of the conditional probability $\mathcal{P}_{aa}(x|0)$ is similar to the probability distribution corresponding to the first excited state of the harmonic oscillator, the maxima of which are indicated with dotted vertical lines in the figure.

In a similar manner one can compare the predictions for the spatial correlations of fermions of different kinds. To this aim, we consider the behaviour of the conditional particle density $\mathcal{D}_{ab}(x|x')$ indicating the density of fermions of kind $a$ at position $x$ conditioned on having found a fermion of kind $b$ at position $x'$. We plot this quantity with $x' = 0$ for the numerical

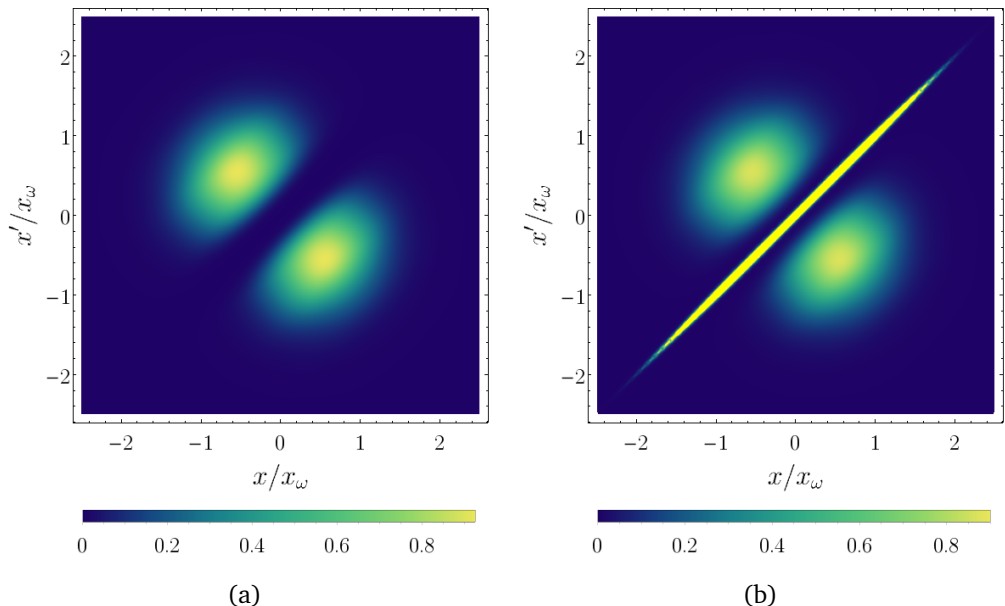

Figure 3: a) Joint density distribution $\mathcal{D}_{aa}(x, x')$ in units of $x_\omega^{-2}$, for two fermions of kind $a$ at positions $x$ and $x'$ simultaneously. b) Joint density distribution $\mathcal{D}_{ab}(x, x')$, in units of $x_\omega^{-2}$, for finding a fermion of kind $a$ at position $x$ and one of kind $b$ at position $x'$ simultaneously. Both densities were obtained from the numerical solution for $\lambda x_\omega = 30$. Details of the calculations are given in Appendices B and C.

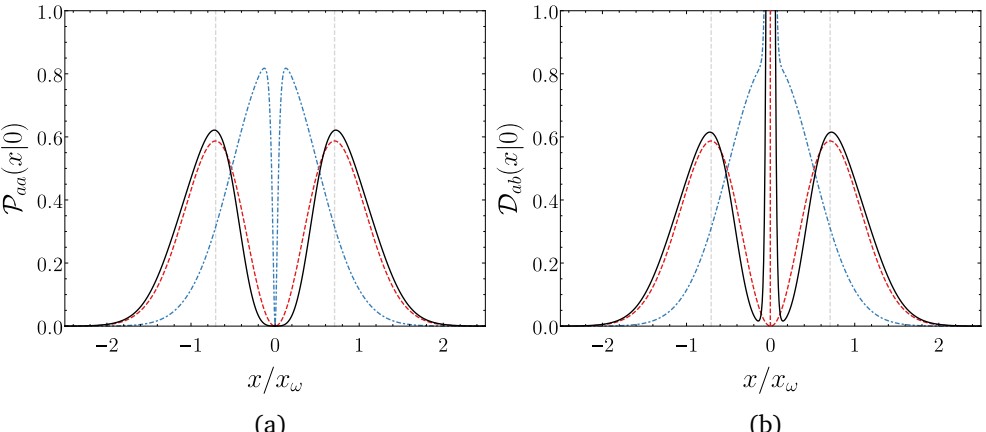

Figure 4: a) Conditional probability $\mathcal{P}_{aa}(x|0)$ to find a fermion of kind $a$ at position $x$ when another fermion was already found at the origin. b) Conditional density $\mathcal{D}_{ab}(x|0)$ indicating the density of fermions of kind $a$ at position $x$ conditioned on having found a fermion of kind $b$ at the origin. In both plots the solid black curve is the numerical result with $\lambda x_\omega = 30$ and $n_{\max} = \lambda x_\omega$. The dashed red curve is the analytical result for the probability obtained for the point-like hard-core boson limit, and the blue dash-dotted line is the probability predicted by the coboson ansatz in Eq. (12) for $N = 2$ and $\lambda x_\omega = 30$. Details of the calculations are given in Appendices B and C. The vertical light-gray lines indicate the positions $\pm x_\omega/\sqrt{2}$, which are the locations of the maxima of the conditional probability for $\lambda \to \infty$.

solution corresponding to $\lambda x_\omega = 30$ in Fig. 4 (b), where we also plot the predictions of the point-like hard-core boson limit and the coboson ansatz for $\lambda x_\omega = 30$. The derivation of the corresponding formulas is shown in Appendix C. All three curves have a narrow peak around the origin, associated with the probability to find a fermion paired with the first one detected (in the limit $\lambda \to \infty$ this peak is a delta function). The curves however differ strongly in the behaviour related with the probability to find the remaining particle of kind $a$. This second contribution to the conditional density has the same shape as $\mathcal{P}_{aa}(x|0)$, and closely resembles the probability distribution for the first excited state of the harmonic oscillator of mass $2m$, a behaviour which is not properly described by the standard coboson ansatz.

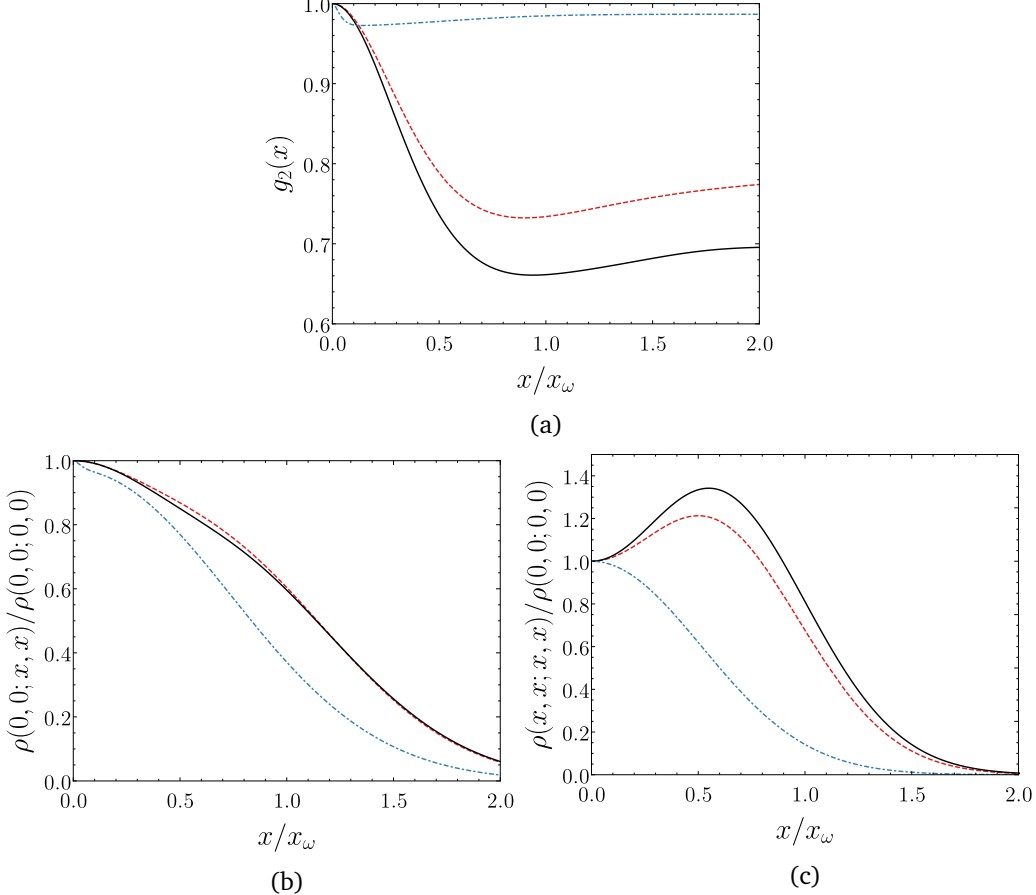

Figure 5: a) Off-diagonal correlation function $g_2(x)$, b) off-diagonal matrix elements and c) diagonal matrix elements of the reduced density matrix $\rho_{ab}$. Black solid lines correspond to numerical results for $\lambda x_\omega = 30$, red dashed ones to point-like hard-core bosons and blue dash-dotted lines correspond to the prediction of the coboson ansatz for $N = 2$ and $\lambda x_\omega = 30$. Details are provided in the main text and in Appendix D. Notice that the vertical axis of subplot (a) does not begin at zero.

Figures 3 and 4 were concerned with density distributions in space, associated with diagonal terms of the system's density matrix in space representation. Figure 5 a) shows in contrast an off-diagonal feature, namely the off-diagonal correlation function [17]:

$$g_2(x) = \frac{\rho_{ab}(0,0;x,x)}{\sqrt{\rho_{ab}(0,0;0,0)\rho_{ab}(x,x;x,x)}}, \tag{27}$$

where $\rho_{ab}$ is the reduced density matrix for two fermions of different kind. The quantity $g_2$ is an indicator of spatial two-particle coherence, and the coboson ansatz predicts a constant

value $g_2(x) = 1$ in the limit of infinite attraction. The numerical results (in black) show that this coherence decays within the typical scale set by the harmonic oscillator, but it stays high for all values with non-negligible particle densities. Nevertheless, the off-diagonal correlation we find is always smaller than the one corresponding to the hard-core limit, depicted in red for comparison. This is not due to a variation in the decay of the spatial coherence, as can be seen in Fig. 5 b). Rather, the difference between our numerical results and the limit $\lambda \to \infty$ is given by a different density profile, since the particle density at the origin is lower for finite $\lambda$ than in the limit of infinite attraction.

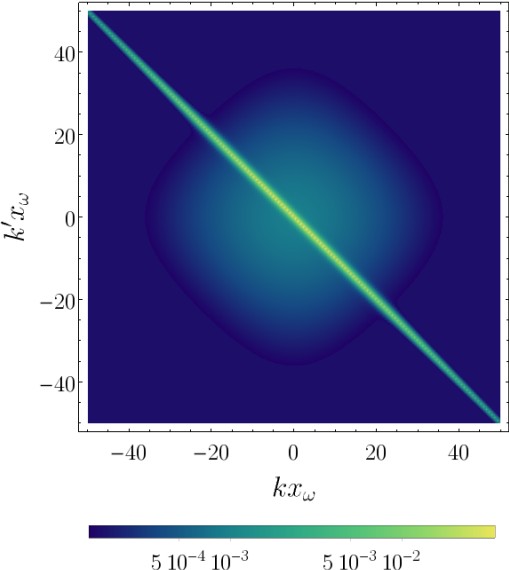

Figure 6: Joint density distribution $\widetilde{\mathcal{D}}_{ab}(k, k')$, in units of $x_\omega^2$, for finding a fermion of kind $a$ with momentum $k$ and one of kind $b$ with momentum $k'$ simultaneously, obtained from the numerical solution for $\lambda x_\omega = 30$. Details are provided in the main text and in Appendix E.

For the same numerically found ground state one can also characterize the properties in momentum space using similar techniques. In Fig. 6 we show the joint probability distribution for fermions of different kinds in momentum space. This plot displays a strong anti-diagonal peak which is the counterpart of the diagonal peak found for the joint probability distribution in position space, shown in Fig. 3 (b). The remaining features of the plot do not ressemble the state of two identical trapped fermions of mass $2m$; this difference in the behaviour of position and momentum is typical of hard-core bosons [7, 32, 47]. The calculation of the joint density in momentum space is similar to the one of $\mathcal{D}_{ab}(x, x')$, but involves a Fourier transform of the coboson basis. The details are explained in Appendix E.

## 5 Summary and conclusions

We have tackled the problem of two identical composite particles, each made of two distinguishable fermions, inside a harmonic trap and with contact attractive interactions between fermions of different species. We explored the strongly bound regime using the coboson formalism to build a compact basis of states, greatly reducing the computational requirements associated with the usual description in terms of single-particle eigenstates.

We have studied the approach of the interaction energy to the limit of infinite attraction, corresponding to point-like hard-core bosons, and we have confirmed that the coboson ansatz

in its standard form does not provide an accurate description of the ground state for any of the interaction strengths within our analysis. Since the energy of the coboson ansatz for $N$ pairs can be approximated from the energy for one and two pairs [23] the coboson ansatz cannot provide a good estimation for the energy of a system made of $N$ pairs. We have also shown that the point-like hard-core boson limit provides a good approximation of the coefficients when writing the ground state in the coboson basis. Furthermore, we have used the numerical results to characterize spatial correlations present in the ground state, both between fermions of different and equal kinds, complementing previous work [17].

The composite-boson procedure presented can be generalized to higher numbers of particles and different forms of the trapping potential. Most importantly, we expect this approach to provide an additional tool to the ones usually applied for the description of experiments involving bosonic Feshbach molecules made of fermionic constituents in quasi one-dimensional settings.

## Acknowledgements

We thank Thomas Busch for his careful reading of the manuscript and valuable comments. E. C. is grateful to Tran Duong Anh-Tai for his suggestions.

**Author contributions**   M. D. J. performed the numerical calculations with support by E. C. and A. P. M. All authors contributed to the derivation of the analytical formulas. C. C. coordinated the project and the writing of the draft.

**Funding information**   The authors acknowledge funding from grant PICT 2017-2583 from ANPCyT (Argentina).

## A   Calculation of Hamiltonian and overlap matrix in position basis

In the limit of very strong interaction, it makes sense to use that the wavefunctions for the center of mass vary over a scale which is much larger than the one for the relative motion. Thus, we start from Eq. (15) for the elements of the overlap matrix, use that all $y_j$ are of the order of $\lambda^{-1}$, and perform a Taylor expansion in these small displacements. The lowest orders give:

$$S_{mn,jk} \simeq \delta_{mj}\delta_{nk} + \delta_{nj}\delta_{mk} - \frac{5}{\lambda}I_{mn,jk} + \frac{7}{8\lambda^3}\int \mathrm{d}x (2\varphi_m\varphi_n\varphi_j'\varphi_k' + \varphi_m\varphi_n'\varphi_j\varphi_k' + \varphi_m'\varphi_n\varphi_j\varphi_k'$$
$$+ \varphi_m\varphi_n'\varphi_j'\varphi_k + \varphi_m'\varphi_n\varphi_j'\varphi_k + 2\varphi_m'\varphi_n'\varphi_j\varphi_k). \quad \text{(A.1)}$$

Here, all functions are evaluated at position $x$, the primes mean that a first derivative must be taken, and $I_{mn,jk}$ is an integral of a product of four single-particle harmonic-oscillator eigenstates:

$$I_{jk,lm} = \int \mathrm{d}x\, \varphi_j(x)\varphi_k(x)\varphi_l(x)\varphi_m(x). \quad \text{(A.2)}$$

These integrals are evaluated using known properties of the Hermite polynomials. In turn, the integrals with derivatives of the eigenfunctions can be written in terms of the elements $I_{mn,jk}$ using the relation:

$$\varphi_n' = \sqrt{\frac{m\omega}{\hbar}}(\sqrt{n}\,\varphi_{n-1} - \sqrt{n+1}\,\varphi_{n+1}), \quad \text{(A.3)}$$

keeping in mind that the $\varphi$ are defined as the eigenfunctions of the harmonic oscillator with mass $2m$.

In the same way one can write an expression for the part of the Hamiltonian involving the commutator, Eq. (18). The dominant contributions give, after some manipulations:

$$\langle v|B_m B_n [V_j^\dagger, B_k^\dagger]|v\rangle \simeq 2\gamma I_{mn,jk}$$

$$-\frac{\gamma}{8\lambda^2}\int dx\,[\varphi_m\varphi_n\varphi_j'\varphi_k' + 3(\varphi_m\varphi_n'\varphi_j\varphi_k' + \varphi_m\varphi_n'\varphi_j'\varphi_k + \varphi_m'\varphi_n\varphi_j\varphi_k' + \varphi_m'\varphi_n\varphi_j'\varphi_k) + 11\varphi_m'\varphi_n'\varphi_j\varphi_k]$$

$$+\frac{\gamma}{128\lambda^4}\int dx\,\{21(\varphi_m\varphi_n'\varphi_j'\varphi_k'' + \varphi_m\varphi_n'\varphi_j''\varphi_k' + \varphi_m'\varphi_n\varphi_j'\varphi_k'' + \varphi_m'\varphi_n\varphi_j''\varphi_k') \tag{A.4}$$

$$+4(\varphi_m\varphi_n''\varphi_j\varphi_k'' + \varphi_m\varphi_n''\varphi_j''\varphi_k + \varphi_m''\varphi_n\varphi_j\varphi_k'' + \varphi_m''\varphi_n\varphi_j''\varphi_k) + 27(\varphi_m\varphi_n''\varphi_j'\varphi_k' + \varphi_m''\varphi_n\varphi_j'\varphi_k')$$

$$-22(\varphi_m'\varphi_n'\varphi_j\varphi_k'' + \varphi_m'\varphi_n'\varphi_j''\varphi_k) + 57\varphi_m''\varphi_n''\varphi_j\varphi_k + \varphi_m\varphi_n\varphi_j''\varphi_k''\},$$

where again all functions are evaluated at position $x$ and the double primes mean that a second derivative must be taken. This formula can be calculated using similar steps as before. Putting this together with the part from $(E_j + E_k)S_{mn,jk}$ we can find a consistent expansion for the Hamiltonian up to this order.

For our numerical calculations, we include the orders reported for $S$ and $H$. One could improve this evaluation by considering higher orders of the Taylor expansion. However, we checked that for the parameter regimes studied the results obtained with these formulas are not significantly altered by excluding the higher order, as can be seen in Fig. 1.

# B  Spatial correlations for two fermions of equal kind

We first consider the joint density distribution for fermions of equal kind:

$$\mathcal{D}_{aa}(x,x') = \langle \psi|\Psi_a^\dagger(x')\Psi_a^\dagger(x)\Psi_a(x)\Psi_a(x')|\psi\rangle, \tag{B.1}$$

of course, taking two fermions of kind $b$ leads in our model to the same result. We note that this definition means that:

$$\iint dx\,dx'\,\mathcal{D}_{aa}(x,x') = 2. \tag{B.2}$$

We now show how we calculate this joint density for the numerically found ground state. Starting from the expansion of the state in the coboson basis, Eq. (11), we find:

$$\mathcal{D}_{aa}(x,x') = \sum_{m\leq n}\sum_{j\leq l} c_{mn}\,c_{jl}\,[J_1^{(jm)}(x)J_1^{(ln)}(x') - J_2^{(jn)}(x,x')J_2^{(ml)}(x,x')]$$

$$+ \text{ same with } n \leftrightarrow m,\ j \leftrightarrow l,\ \text{and } \{n,l\} \leftrightarrow \{m,j\}. \tag{B.3}$$

The result for the standard coboson ansatz corresponds to setting all $c_{jl}$ to zero except for $c_{00}$. In the formula above we have introduced auxiliary integrals given by:

$$J_1^{(jm)}(x) = \int dx'\,\psi_j(x,x')\psi_m(x,x'), \tag{B.4}$$

and

$$J_2^{(jm)}(x,x') = \int dx''\,\psi_j(x,x'')\psi_m(x',x''). \tag{B.5}$$

The integrals $J_2$ account for fermion-exchange terms and are negligible unless $x$ and $x'$ are close together within a distance of order $1/\lambda$.

From these formulas one can recover the vanishing of the conditional probability for $x = x'$ for arbitrary states. For the limit $\lambda \to \infty$, the joint density $\mathcal{D}_{aa}$ tends to the expression given in Eq. (21) which was calculated from the ground state of two point-like hard-core bosons. In the limit of very large but finite attraction, one can resort to a Taylor expansion for the calculation of the integrals, in the same spirit as the calculations in Appendix A. For $J_1$ we obtain:

$$J_1^{(jm)}(x) \simeq \varphi_j \varphi_m + \frac{1}{16\lambda^2}[2\varphi_j'\varphi_m' + \varphi_j''\varphi_m + \varphi_j\varphi_m'']$$
$$+ \frac{1}{256\lambda^4}[6\varphi_j''\varphi_m'' + 4\varphi_j^{(3)}\varphi_m' + 4\varphi_j'\varphi_m^{(3)} + \varphi_j^{(4)}\varphi_n + \varphi_j\varphi_m^{(4)}]. \quad \text{(B.6)}$$

Here, all functions are evaluated at position $x$ and we are using primes (double primes) over the functions to denote derivatives (second derivatives), whereas derivatives of higher order are indicated with superindices between parenthesis. We remind the reader that the $\varphi_n$ indicate the oscillator eigenfunctions for mass $2m$.

The integral for $J_2$ can be expanded as:

$$J_2^{(jm)}(x,x') \simeq e^{-\lambda|x-x'|}\Big\{ \varphi_j(x)\varphi_m(x')(1 + \lambda|x-x'|)$$
$$- \frac{1}{4\lambda}\Big[\varphi_j'(x)\varphi_m(x') - \varphi_j(x)\varphi_m'(x')]\lambda(x-x')(1 + \lambda|x-x'|)\Big]$$
$$+ \frac{1}{6\lambda^2}\Big[\frac{1}{4}\varphi_j'(x)\varphi_m'(x')\Big(3 + 3\lambda|x-x'| - \lambda^3|x-x'|^3\Big)$$
$$+ \frac{1}{8}\Big(\varphi_j''(x)\varphi_m(x') + \varphi_j(x)\varphi_m''(x')\Big)\Big(3 + 3\lambda|x-x'| + 3\lambda^2(x-x')^2 + 2\lambda^3|x-x'|^3\Big)\Big]\Big\}. \quad \text{(B.7)}$$

From the joint density $\mathcal{D}_{aa}(x,x')$ one can also calculate the conditional probability $\mathcal{P}_{aa}(x|x')$ of finding a particle of kind $a$ at position $x$ when another of the same kind was found at position $x'$. This can be computed from:

$$\mathcal{P}_{aa}(x|x') = \frac{\mathcal{D}_{aa}(x,x')}{\langle\psi|\Psi_a^\dagger(x')\Psi_a(x')|\psi\rangle}, \quad \text{(B.8)}$$

so that:

$$\int dx\, \mathcal{P}_{aa}(x|x') = 1 \quad \forall\, x'. \quad \text{(B.9)}$$

For the limit $\lambda \to \infty$, the conditional probability $\mathcal{P}_{aa}$ tends to the expression given in Eq. (22) calculated from the ground state of two point-like hard-core bosons.

On the other hand, the standard coboson ansatz predicts for $\lambda \to \infty$ a behaviour of the form:

$$\mathcal{D}_{aa}(x,x') = \begin{cases} 2\varphi_0(x)^2\varphi_0(x')^2 & \text{if } x \neq x', \\ 0 & \text{if } x = x', \end{cases} \quad \text{(B.10)}$$

so that

$$\mathcal{P}_{aa}(x|x') = \begin{cases} \varphi_0(x)^2 & \text{if } x \neq x', \\ 0 & \text{if } x = x'. \end{cases} \quad \text{(B.11)}$$

## C  Spatial correlations for fermions of different kinds

We now calculate spatial correlations between fermions of different kinds. In particular, we are interested in the joint particle density

$$\mathcal{D}_{ab}(x,x') = 4\rho_{ab}(x_a, x_b; x_a, x_b). \quad \text{(C.1)}$$

Here, $\rho_{ab}$ is the reduced density matrix of two different fermions in position basis and is given by [17]:

$$\rho_{ab}(x_a, x_b; x_a', x_b') = \frac{1}{4}\langle\psi|\Psi_a^\dagger(x_a)\Psi_b^\dagger(x_b)\Psi_b(x_b')\Psi_a(x_a')|\psi\rangle. \tag{C.2}$$

In the following we proceed to the calculation of the joint density for the general numerical solution. Replacing the expansion of the state in the coboson basis leads to:

$$\begin{aligned}
\mathcal{D}_{ab}(x, x') = \sum_{m\leq n}\sum_{j\leq l} c_{mn}\, c_{jl} &\left\{\left[\delta_{nl}\psi_m(x, x')\psi_j(x, x') + J_1^{(ln)}(x)J_1^{(jm)}(x')\right.\right. \\
&\left.- \psi_m(x, x')J_3^{(jl|n)}(x, x') - \psi_j(x, x')J_3^{(mn|l)}(x, x')\right] \\
&\left.+ \text{same with } n \leftrightarrow m,\ j \leftrightarrow l,\ \text{and } \{n, l\} \leftrightarrow \{m, j\}\right\}. \tag{C.3}
\end{aligned}$$

Here, the $J_1$ are given in Eq. (B.4), and the $J_3$ contain interference terms given by:

$$J_3^{(jl|n)}(x, x') = \int dy\, dy'\, \psi_j(x, x-y)\,\psi_l(x'+y', x')\,\psi_n(x-y, x'+y'). \tag{C.4}$$

Again, the result for the coboson ansatz is found setting all coefficients $c_{jl}$ to zero except for $c_{00}$.

Resorting to the Taylor expansion $J_3^{(jl|n)}(x, x')$ can be approximated by

$$\begin{aligned}
J_3^{(jl|n)}(x, x') \simeq e^{-\lambda|x-x'|}&\left\{\frac{1}{2\sqrt{\lambda}}\varphi_j(x)\varphi_l(x')\varphi_n(x)\left(\lambda^2(x-x')^2 + 3\lambda|x-x'| + 3\right)\right. \\
&+ \frac{1}{12\sqrt{\lambda}}\left[-\varphi_j'(x)\varphi_l(x')\varphi_n(x) + \varphi_j(x)\varphi_l'(x')\varphi_n(x) - 3\varphi_j(x)\varphi_l(x')\varphi_n'(x)\right] \\
&\qquad\times (x-x')\left(\lambda^2(x-x')^2 + 3\lambda|x-x'| + 3\right) \\
&+ \frac{1}{24\lambda^{5/2}}\left[-\frac{1}{4}\varphi_j'(x)\varphi_l'(x')\varphi_n(x) - \frac{3}{4}\varphi_j(x)\varphi_l'(x')\varphi_n'(x) + \frac{1}{4}\varphi_j'(x)\varphi_l(x')\varphi_n'(x)\right. \\
&\left.+ \frac{1}{2}\varphi_j(x)\varphi_l(x')\varphi_n''(x)\right] \times \left(\lambda^4(x-x')^4 + 2\lambda^3|x-x'|^3 - 3\lambda^2(x-x')^2 - 15\lambda|x-x'| - 15\right) \\
&+ \frac{1}{12\lambda^{5/2}}\left[\frac{1}{2}\varphi_j'(x)\varphi_l(x')\varphi_n'(x) + \frac{1}{8}\varphi_j''(x)\varphi_l(x')\varphi_n(x) + \frac{1}{8}\varphi_j(x)\varphi_l''(x')\varphi_n(x)\right. \\
&\left.\left.+ \frac{5}{8}\varphi_j(x)\varphi_l(x')\varphi_n''(x)\right] \times \left(\lambda^4(x-x')^4 + 4\lambda^3|x-x'|^3 + 9\lambda^2(x-x')^2 + 15\lambda|x-x'| + 15\right)\right\}. \tag{C.5}
\end{aligned}$$

We note that just as in $\mathcal{P}_{aa}$, the terms with $J_1$ are the only ones that are non-negligible when $x$ and $x'$ are at a distance much larger than $1/\lambda$. Thus, $\mathcal{P}_{aa}$ and $\mathcal{D}_{ab}$ behave in the same way for $e^{-\lambda|x-x'|} \ll 1$, corresponding to detection of particles in different bound pairs. In the opposite limit of $x$ close to $x'$, $\mathcal{D}_{ab}$ has a peak of width $1/\lambda$ corresponding to detection of the particle forming a pair with the fermion detected at $x'$.

We now calculate a quantity analogue to $\mathcal{P}_{aa}(x|x')$ but applying to fermions of different kinds. In particular, we wish to calculate the conditional probability $\mathcal{P}_{ab}(x|x')$ of finding a fermion of kind $a$ at position $x$ conditioned on having found a fermion of kind $b$ at position $x'$. This, however, is trickier because after the detection of one fermion of kind $b$ there are two remaining identical fermions of kind $a$.

Thus, we choose to work with a conditional particle density $\mathcal{D}_{ab}(x|x')$ indicating the density of fermions of kind $a$ at position $x$ conditioned on having found a fermion of kind $b$ at position $x'$. Since two identical fermions can never be found at the same place, this quantity is related with the conditional probability $\mathcal{P}_{ab}$, but its interpretation is more straightforward and,

in contrast with a probability, $\mathcal{D}_{ab}$ must be normalized to 2. More precisely, the conditional particle density is given by:

$$\mathcal{D}_{ab}(x|x') = \frac{\mathcal{D}_{ab}(x,x')}{\langle\psi|\Psi_b^\dagger(x')\Psi_b(x')|\psi\rangle} \,. \tag{C.6}$$

such that

$$\int dx \, \mathcal{D}_{ab}(x|x') = 2 \quad \forall \, x'. \tag{C.7}$$

For infinite attraction, it holds that $\mathcal{D}_{ab}(x|x') = \mathcal{P}_{aa}(x|x') + \delta(x-x')$. For the coboson ansatz, in the limit $\lambda \to \infty$ one has $\mathcal{D}_{ab}(x,x') = 2\varphi_0(x)^2[\varphi_0(x')^2 + \delta(x-x')]$ and accordingly $\mathcal{D}_{ab}(x|x') = \varphi_0(x')^2 + \delta(x-x')$.

## D  Off-diagonal correlation parameter

Here we provide the expression for the off-diagonal correlation parameter $g_2(x)$ defined in Eq. (27). The diagonal matrix elements appearing in the denominator are particular instances of the calculation in the previous section, so that one can use Eq. (C.3) evaluated for $x' = x$. For the off-diagonal part, we plug the decomposition of the state in the coboson basis and apply (anti)commutators as in the previous sections.

$$\begin{aligned}
\rho_{ab}(0,0;x,x) \simeq \frac{1}{4}\sum_{m\leq n}\sum_{j\leq l} c_{mn}\,c_{jl} \Big\{&\Big[\psi_j(x,x)\psi_m(0,0)\delta_{nl} + J_2^{(ln)}(x,0)J_2^{(jm)}(x,0) \\
&- \psi_j(x,x)J_4^{(mn|l)}(0) - \psi_m(0,0)J_4^{(jl|n)}(x)\Big] \\
&+ \text{same with } n \leftrightarrow m, \; j \leftrightarrow l, \text{ and } \{n,l\} \leftrightarrow \{m,j\}\Big\},
\end{aligned} \tag{D.1}$$

where we have introduced a new integral expression:

$$J_4^{(mn|l)}(x) = \int dy\,dy'\, \psi_m(y,x)\psi_n(x,y')\psi_l(y,y'), \tag{D.2}$$

that can be Taylor-expanded as follows:

$$\begin{aligned}
J_4^{(mn|l)} \simeq &\frac{3}{2\sqrt{\lambda}}\varphi_m\varphi_n\varphi_l \\
&+ \frac{5}{4\lambda^{5/2}}[\varphi_m'\varphi_n'\varphi_l + 3\varphi_m'\varphi_n\varphi_l' + 3\varphi_m\varphi_n'\varphi_l' + \varphi_m''\varphi_n\varphi_l + \varphi_m\varphi_n''\varphi_l + 3\varphi_m\varphi_n\varphi_l''],
\end{aligned} \tag{D.3}$$

with all functions evaluated at the same position.

In the limit of infinite attraction the form of $g_2$ can be calculated using the point-like hard-core boson solution. This gives:

$$g_{2-\text{hc}}(x) = \frac{\frac{4x}{\sqrt{2\pi}} + x_\omega \text{erfc}(\sqrt{2}\,x/x_\omega)}{\sqrt{4x^2 + x_\omega^2}}, \tag{D.4}$$

where "erfc" is the complementary error function. This is a quite flat behaviour for $g_2$, but still clearly different from the totally flat profile, $g_2(x) = 1 \; \forall \, x$, that is obtained from the standard coboson ansatz for $\lambda \to \infty$.

# E  Correlations in momentum space

One can easily extend the results from the previous appendices to momentum space. In order to do this, we resort to the expresion of the coboson wavefunctions in momentum space:

$$\widetilde{\psi}_n(k_1, k_2) = \sqrt{\frac{2}{\pi\lambda}} \frac{x_\omega e^{-i\pi n/2}}{1 + \left(\frac{k_1 - k_2}{2\lambda}\right)^2} \varphi_n[x_\omega^2(k_1 + k_2)], \tag{E.1}$$

which is just the Fourier transform of Eq. (4). These functions are of order $\sqrt{x_\omega/\lambda}$ and decay in a scale of order $\lambda$ for the relative momentum $(k_1 - k_2)/2$ and of order $\sqrt{n}/x_\omega$ for the center-of-mass momentum $k_1 + k_2$.

From this expression one can derive formulas for the correlations in momentum space following similar steps as before. One should only keep in mind that, in contrast to position space, the wavefunctions in momentum space are complex. In particular, we find for the momentum correlations between fermions of different kinds an equation which is analogous to Eq. (C.3):

$$\widetilde{\mathcal{D}}_{ab}(k, k') = \sum_{m \leq n} \sum_{j \leq l} c_{mn} c_{jl} \left\{ \left[ \delta_{nl} \widetilde{\psi}_m^*(k, k') \widetilde{\psi}_j(k, k') + \widetilde{J}_1^{(nl)}(k) \widetilde{J}_1^{(mj)}(k') \right. \right.$$
$$\left. - \widetilde{\psi}_m^*(k, k') \widetilde{J}_3^{(jl|n)}(k, k') - \widetilde{\psi}_j(k, k')[\widetilde{J}_3^{(mn|l)}(k, k')]^* \right] \tag{E.2}$$
$$\left. + \text{same with } n \leftrightarrow m, \, j \leftrightarrow l, \text{ and } \{n, l\} \leftrightarrow \{m, j\} \right\}.$$

Here, the asterisk denotes a complex conjugation and we have defined:

$$\widetilde{J}_1^{(nl)}(k) = \int dk' \widetilde{\psi}_n^*(k, k') \widetilde{\psi}_l(k, k'), \tag{E.3}$$

and

$$\widetilde{J}_3^{(jl|n)}(k, k') = \int dq \, dq' \widetilde{\psi}_j(k, q') \widetilde{\psi}_l(q, k') \widetilde{\psi}_n^*(q, q'). \tag{E.4}$$

Replacing the form of the wavefunctions in momentum space one finds the integral expression:

$$\widetilde{J}_1^{(nl)}(k) = \frac{2x_\omega^2}{\pi\lambda} e^{i\pi(n-l)/2} \int dk' \frac{\varphi_n(x_\omega^2 k') \varphi_l(x_\omega^2 k')}{\left[1 + \left(\frac{k'-2k}{2\lambda}\right)^2\right]^2}. \tag{E.5}$$

Taking into account the restriction on the values of $n, l$ within our basis, one can perform a Taylor expansion in $k'/\lambda$ in the expression above. We stress that the values of $k$ cannot be assumed to be much smaller than $\lambda$, since $\lambda$ is indeed the typical scale for the relative momentum. In this way we obtain:

$$\widetilde{J}_1^{(nl)}(k) \simeq \frac{2x_\omega^2}{\pi\lambda} e^{i\pi(n-l)/2} \int dk' \varphi_n(x_\omega^2 k') \varphi_l(x_\omega^2 k')$$
$$\times \left[ \frac{1}{\left(\frac{k^2}{\lambda^2} + 1\right)^2} + \frac{2kk'}{\lambda^2 \left(\frac{k^2}{\lambda^2} + 1\right)^3} - \frac{(1 - \frac{5k^2}{\lambda^2})k'^2}{2\lambda^2 \left(\frac{k^2}{\lambda^2} + 1\right)^4} \right], \tag{E.6}$$

which can be evaluated using properties of the Hermite polynomials.

The integrals for $\widetilde{J}_3$ can be cast in the form:

$$
\widetilde{J}_3^{(jl|n)}(k, k') = \left(\frac{2x_\omega^2}{\pi\lambda}\right)^{3/2} e^{-i\pi(j+l-n)/2}
$$
$$
\times \int dq\, dq' \frac{\varphi_j(x_\omega^2 q')\varphi_l(x_\omega^2 q)\varphi_n[x_\omega^2(q+q'-k-k')]}{\left[1+(\frac{q'-2k}{2\lambda})^2\right]\left[1+(\frac{q-2k'}{2\lambda})^2\right]\left[1+(\frac{k-k'+q-q'}{2\lambda})^2\right]}. \tag{E.7}
$$

Performing a Taylor expansion here is justified for the $q$, $q'$ divided by $\lambda$ in the denominator, but not for the same variables inside the wavefunction $\varphi$. This makes this calculation much more involved. The Taylor expansion of the denominator gives:

$$
\widetilde{J}_3^{(jl|n)}(k, k') \simeq \left(\frac{2x_\omega^2}{\pi\lambda}\right)^{3/2} e^{-i\pi(j+l-n)/2} \int dq\, dq'\, \varphi_j(x_\omega^2 q')\varphi_l(x_\omega^2 q)\varphi_n[x_\omega^2(q+q'-k-k')]
$$
$$
\times \left[ \frac{1}{\left(\frac{k^2}{\lambda^2}+1\right)^2\left(\frac{k'^2}{\lambda^2}+1\right)^2\left(\frac{(k-k')^2}{4\lambda^2}+1\right)^2} \right.
$$
$$
\left. + \frac{(k-3k')\left(\frac{k'(k-k')}{2\lambda^2}-1\right)}{\lambda^2\left(\frac{k^2}{\lambda^2}+1\right)^2\left(\frac{k'^2}{\lambda^2}+1\right)^3\left(\frac{(k-k')^2}{4\lambda^2}+1\right)^3}q + \frac{(3k-k')\left(\frac{k(k-k')}{2\lambda^2}+1\right)}{\lambda^2\left(\frac{k^2}{\lambda^2}+1\right)^3\left(\frac{k'^2}{\lambda^2}+1\right)^2\left(\frac{(k-k')^2}{4\lambda^2}+1\right)^3}q' \right], \tag{E.8}
$$

Here one is still left with a non-trivial integral in $q, q'$. This can be solved using the decomposition formula

$$
\varphi_n(x+y) = \sum_{i,j=0}^{\infty} A_{ij|n}\varphi_i(x)\varphi_j(y), \tag{E.9}
$$

where the coefficients

$$
A_{ij|n} = \int\int \varphi_n(x+y)\varphi_i(x)\varphi_j(y)\,dx\,dy, \tag{E.10}
$$

can be evaluated using properties of the Hermite polynomials. For numerical evaluation this summation must be truncated. Performing this one up to $i, j = 100$ good approximations are obtained. Then we are left with terms similar to those found in the calculation for $\widetilde{J}_1$.

The first term in Eq. (E.2) contains contributions of order $x_\omega/\lambda$ which decay in a scale of order $\lambda$ for the relative momentum $(k-k')/2$ and of order $1/x_\omega$ for the center-of-mass momentum $k+k'$. The second term, involving $\widetilde{J}_1$, contains contributions of order $1/\lambda^2$ which decay on a scale of order $\lambda$ for $k$ and $k'$ separately. We note that this contribution is broad and has a Lorentzian decay, whereas the decay of the contributions in the first term is Gaussian for the center of mass. Thus, they may be of the same order depending on the point where they are evaluated. In any case, the dominant feature is the anti-diagonal resulting from the first term. The remaining terms, containing $\widetilde{J}_3$, have a similar behaviour as the first (i.e. with a strong anti-diagonal) but are one order smaller in $1/(\lambda x_\omega)$, which justifies using an expansion for $\widetilde{J}_3$ to a lower order than for $\widetilde{J}_1$.

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
