# Peer review of "Composite-boson formalism applied to strongly bound fermion pairs in a one-dimensional trap"

_SciPost Physics Core, doi:SciPost Phys. Core 6, 012 (2023)_

## Round 1 · Referee Report · Anonymous (Referee 1) · 2022-9-14

Strengths

Science preformed and presented in the paper is clear, accurate and sound.

Weaknesses

Motivation is lacking, and discussion/connection to the broader community is limited.

Report

The authors find the ground state of 4 interacting fermions in 1D. Two fermions are of type "a" and the other two are of type "b", and fermions of different species are strongly attractive and form a tightly bound state. Such a tightly bound state allows for analytic approximations, and an hilbert space truncation that increases numerical efficiency. The results presented are sound and representative of good science, and the presentation is accurate and clear. I therefore recommend publication after a the minor changes are made as described below.

In addition, the authors can improve the quality of the paper by addressing the the following concerns:

1) The paper appears to fit in a broader context then discussed in the introduction. Specifically, they are studying the ground state of 4 fermions, and the methods they use should be similar to related calculations in quantum chemistry. 2) It seems like there should be integrable models of the form similar to the ones studied by the authors such as those discussed in Guan et. al. Rev. Mod. Phys. 85, 1633 (2013). A discussion on this would be helpful to put the paper in a broader context. 3) What do the authors think for N>2 states? Will similar physics hold in a many body limit? 4) Adding the coboson ansatz line to fig 5a could be helpful even though it is trivial.

Requested changes

1) The authors refer to a true ground state |GS>. I think this is the ground state computed using the basis in equation 10, but it is not clear from the text. The authors should clarify this. 2) The extent to which |GS> is the "true" ground state depends on the validity of approximation in eq 4. This should be clarified, and the conditions should be specified for when this approximation is valid. 3) Specify N=2 for the coboson ansatz when discussing equation 12 4) The authors should explain why the infinite interaction strength approximation seems to work so well in fig 2, but less so in fig 1.

Also a few issues on language: 1) This is a missing period between eq 11 and 12 2) In eq 14, the sum over j is confusing because j is already a fixed index. 3) There is a "this" that refers to equation 9, but I think the authors intended it to refer to something else. 4)"quarctic" above equation 15 5) The phrase "one-pair eigenstate" at the bottom of page 2 was not clear on first reading. It is also inconsistent with "single-pair" basis which is also used. 6)The sentence above equation 8 is confusing. On first reading it seemed to suggest the index n was indexing the excited internal states. 7) Above equation 15, there is a cryptic sentence about the one body term not playing a role in the energy. This is not true, and the statement in the "( ...)" is not clear enough to explain. 8) Add more context to equation 18. As is, it was presented out of no where, was it proved in reference 7? 9) The authors should explain why this wave function has interaction energy \hbar \omega, or give more justification then is present. 10) In the last paragraph on page 7, the authors refer to "the coefficients". Which coefficients are they referring to? 11) The second paragraph of page 9 refers to N=2 without context as to what N is.

---

## Round 1 · Referee Report · Anonymous (Referee 2) · 2022-10-10

Strengths

  • interesting (semi-) analytical ansatz
  • clearly written

Weaknesses

  • very limited accessible system size
  • larger systems numerically easily accessible using other established methods
  • significance and impact of results remain unclear

Report

The work addresses a problem of four particles in one dimension, using a newly proposed semi-analytical ansatz wavefunction. The paper is clearly written, and the presented science is sound and well explained.
However, the main problem I see is that the type of problem addressed here, even with significantly larger particle numbers, can be addressed using established numerical approaches, such as for example diffusion Monte Carlo methods or the multi configuration self-consistent field method. Hence I find it difficult to understand how the present work significantly advances the current knowledge of the field, and which important questions the paper addresses (being restricted to just four particles - can/did the authors go beyond N=4?). Before the authors convince me of their motivation and the significance of their results, I cannot conclude that the manuscript meets the required acceptance criteria.

---

## Round 2 · Referee Report · Anonymous (Referee 3) · 2022-11-3

Report

I appreciate the effort of the authors to better motivate their approach. As I already concluded in my first report, the work is clearly written and I strongly believe it is technically sound. I find the motivation explained in the replies to the first reports, to extend this method to more complicated many-body systems, well justified and a worthy goal. Hence I can recommend publication in SciPost Physics Core.

---

## Round 2 · Referee Report · Anonymous (Referee 4) · 2022-11-8

Report

Authors have properly addressed my concerns and I recommend the article for publication.

A small typo:
In the sentence "In this respect, our method is related with the perturbative approach in [35]", "with" should be "to"

---

## Round 2 · Author Response

Response to Referee Report 1:

We are pleased to read that the referee has found our work clear, accurate and sound, and we are grateful for the careful reading and the many suggestions and observations that have significantly improved the presentation and clarity of our manuscript. In the following we address each of the comments and requests individually.

Comments in the main report:

1) The paper appears to fit in a broader context then discussed in the introduction. Specifically, they are studying the ground state of 4 fermions, and the methods they use should be similar to related calculations in quantum chemistry.

We agree with the reviewer that the broader context of our work was not clear in our previous version. Although the field is way too broad to be properly covered in an article which is not a review, we have made an effort to succinctly include relevant references which can help the reader better frame our research and find more information about different approaches to various regimes and related models. We note that our former Ref. [8] indeed referred to quantum-chemical treatments, but we have now included citations to many more numerical and analytical approaches.

2) It seems like there should be integrable models of the form similar to the ones studied by the authors such as those discussed in Guan et. al. Rev. Mod. Phys. 85, 1633 (2013). A discussion on this would be helpful to put the paper in a broader context.

We agree with the reviewer and we have largely expanded our bibliography to include some of this work, as well as reviews that can provide even more references for the interested reader.

3) What do the authors think for N>2 states? Will similar physics hold in a many body limit?

Indeed, we expect the case N=2 to be representative of the behaviour for larger particle numbers. In particular, the fermionization procedure for hard-core bosons corresponding to the limit of infinite attraction is general. An ansatz based on this limit as suggested after Fig. 1, i.e. simply replacing the coefficients of the variational coboson basis by the asymptotic values for the given N, can be applied for larger particle numbers. Also the general approach can be used for larger N, although the calculations will involve more integrals. Since the careful treatment of the case of arbitrary N is beyond the scope of this work, we have limited ourselves to mention this possible extension both in the introduction and the conclusion.

4) Adding the coboson ansatz line to fig 5a could be helpful even though it is trivial.

We have added to Fig. 5 all curves corresponding to the coboson ansatz. Actually, the previous version of the manuscript had a mistake in the writing: it was stated that the coboson ansatz predicts a value of 1 for the off-diagonal correlator, but this is only true for infinite attraction. For finite but strong attraction, g2 is close to 1 but contains corrections of order (λxω)^-1. We have corrected this mistake and we thank the reviewer for making us notice this problem.

Requested changes:

1) The authors refer to a true ground state |GS>. I think this is the ground state computed using the basis in equation 10, but it is not clear from the text. The authors should clarify this.

System Message: WARNING/2 (<string>, line 27); backlink

Inline substitution_reference start-string without end-string.

The reviewer is right and we have added a sentence after Eq. (12) which reads:

“where the true ground state |GS〉 is approximated numerically using the coboson basis given in Eq. (10) for two-pairs (N = 2).”

System Message: WARNING/2 (<string>, line 31); backlink

Inline substitution_reference start-string without end-string.

2) The extent to which |GS> is the "true" ground state depends on the validity of approximation in eq 4. This should be clarified, and the conditions should be specified for when this approximation is valid.

System Message: WARNING/2 (<string>, line 33); backlink

Inline substitution_reference start-string without end-string.

The reviewer is right and we have added a sentence at the end of Section 2.1 addressing this point, which reads:

“It is also important to note that since Eq. (2) and therefore Eq. (4) are valid for λ xω ⪆ 5 all of our results rely on this condition [30].”

We have also provided a new reference showing the validity of the approximation in this regime.

3) Specify N=2 for the coboson ansatz when discussing equation 12

We have added this clarification in the sentence mentioned in the previous item 1).

4) The authors should explain why the infinite interaction strength approximation seems to work so well in fig 2, but less so in fig 1.

We think that plotting the interaction energy between pairs instead of the total energy is what makes this point confusing. We have added before Fig. 1 the sentence:

“Notice that the difference between the numerical ∆E obtained for λxω ≃ 200 and the asymptotic value ħω presented in Fig. 1 (a) is of about 4%, whereas the binding energy for this case is so large that ∆E is five orders of magnitude smaller than the total energy.”

We hope this clarifies the point. We thank the reviewer for pointing out this issue.

Issues on language:

1) This is a missing period between eq 11 and 12

We have fixed this typo, and we thank the reviewer for spotting it and letting us know.

2) In eq 14, the sum over j is confusing because j is already a fixed index.

The reviewer is right. We have changed the index in the sum from j to l.

3) There is a "this" that refers to equation 9, but I think the authors intended it to refer to something else.

We are sorry if we have not been clear in the wording. The text after Eq. (9) now reads:

“The inequality in Eq. (8) stresses once more the fact that our restricted basis is only appropriate for strong attraction, when the size of each bound pair is very small compared with the spatial scale of the trap and thus λxω is large”.

4) "quarctic" above equation 15

We have fixed this misspelling, we thank the reviewer for pointing it out.

5) The phrase "one-pair eigenstate" at the bottom of page 2 was not clear on first reading. It is also inconsistent with "single-pair" basis which is also used.

We have made this consistent using “single-pair” in all cases.

6) The sentence above equation 8 is confusing. On first reading it seemed to suggest the index n was indexing the excited internal states.

We thank the reviewer for this observation. We have clarified this part of the text which now reads:

“For consistency, neglecting states where the internal motion is excited implies also a truncation in the center-of-mass states, so that the basis includes all single-pair eigenstates up to a certain energy cutoff. In particular, we keep only states where the index n associated with the center-of-mass motion is such that the excited internal states are well above the energy scales considered, i.e.:”

We have also added an intermediate step in Eq. (8) to clarify the interplay between the energies involved.

7) Above equation 15, there is a cryptic sentence about the one body term not playing a role in the energy. This is not true, and the statement in the "( ...)" is not clear enough to explain.

We did not mean to imply that the one-body term did not play a role in the energy, but we were making heavy use of previous results in the coboson literature, and we are sorry that this was not clear enough. We have removed the cryptic sentence and introduced a new sentence in the paragraph after the present Eq. (17), which reads:

“Notice that when using the coboson formalism the one-body term which contains the kinetic energy and trap potential is absorbed by quantities that were calculated when solving the single-pair case (first term on the right-hand-side in the above equation).“

We hope that the text has become more reader-friendly with this clarification.

8) Add more context to equation 18. As is, it was presented out of no where, was it proved in reference 7?

Ref. [7] does not provide the explicit form of the former Eq. (18) - now Eq. (20) - but rather the general procedure involving fermionization. We have now added an intermediate equation, present Eq. (19), for the ground state for two fermions of mass 2m, which we hope clarifies the point.

9) The authors should explain why this wave function has interaction energy hbar omega, or give more justification then is present.

We have expanded the paragraph to explain the derivation better. The present paragraph reads:

“From these expressions we can calculate all properties of the ground state for λ → ∞. For instance, the asymptotic ground-state energy, excluding the binding energy Eγ of each pair, is found to be given by the sum of the two lowest energies of the harmonic oscillator. Thus, the total ground-state energy for very large λ is approximately 2Eγ + 2ħω. We can define an effective interaction energy between pairs as ∆E = E2 − 2E1, where EN is the ground-state energy of N = 1, 2 pairs. Considering that a single pair has a ground-state energy of Eγ + ħω/2, we then obtain an effective interaction energy which for very large attraction approaches ∆E = ħω.”

10) In the last paragraph on page 7, the authors refer to "the coefficients". Which coefficients are they referring to?

We have clarified this point referring to the corresponding equation. We have also added a note in Fig. 2 explaining that a single index k has been used to label the combination m,n.

11) The second paragraph of page 9 refers to N=2 without context as to what N is.

Here N was referring to the number of pairs, but we removed that part of the sentence since the number of pairs was fixed throughout the whole section.

Response to Referee Report 2:

We are glad that the reviewer finds our work sound and well explained, and we are sorry that we have been unable to properly convey the motivation, goals and scope of our study. We hope the present version of our manuscript addresses these issues. With this purpose, we have largely expanded the bibliographical references to provide a broader context to our analysis, and we have more clearly stated the specific objectives of our present work and the virtues of our approach, together with possible outlooks. We agree that there are other numerical techniques that can be applied to the same problem and we have now included more information about them. However, we think that this does not diminish the value of our contribution. First of all, because our method is numerically very economical, particularly in a regime in which most methods are highly costly. But most importantly, because we are convinced that new approaches to the same problems can bring new insights and also be useful to answer different kinds of questions. The coboson mindset, in particular, brought to light the relation between important models in condensed matter and key concepts of quantum information, by showing that a large enough amount of entanglement between the constituent parts ensures the bosonic behaviour of composite particles. We believe that this bridge is interesting and deserves further exploration, and we consider our present work as one step in this direction.

---

## Round 2 · List of Changes

- We have largely expanded the number of bibliographical references, including a more complete description of historical developments in the field and tools to approach the model that we study in different regimes.

- We have better stated our goals, the scope of our work, and its connection with the existing literature.

- We have improved the formatting of the figures.

- We have added the coboson ansatz to Fig. 5 and clarified that it predicts a g2 equal to 1 only for infinite attraction.

- We have made several writing corrections and clarifications as requested by Referee 1.

- We have better explained the derivations and results involving the fermionization procedure.

- We have added acknowledgements to Thomas Busch and Tran Duong Anh-Tai.

---

## Editorial Decision

published